# Adaptive thermal plasticity enhances sperm and egg performance in a model insect

**Ramakrishnan Vasudeva, Andreas Sutter, Kris Sales, Matthew E Dickinson, Alyson J Lumley, Matthew JG Gage\***

School of Biological Sciences, University of East Anglia, Norwich, United Kingdom

**Abstract** Rising and more variable global temperatures pose a challenge for biodiversity, with reproduction and fertility being especially sensitive to heat. Here, we assessed the potential for thermal adaptation in sperm and egg function using *Tribolium* flour beetles, a warm-temperate-tropical insect model. Following temperature increases through adult development, we found opposing gamete responses, with males producing shorter sperm and females laying larger eggs. Importantly, this gamete phenotypic plasticity was adaptive: thermal translocation experiments showed that both sperm and eggs produced in warmer conditions had superior reproductive performance in warmer environments, and *vice versa* for cooler production conditions and reproductive environments. In warmer environments, gamete plasticity enabled males to double their reproductive success, and females could increase offspring production by one-third. Our results reveal exciting potential for sensitive but vital traits within reproduction to handle increasing and more variable thermal regimes in the natural environment.
DOI: https://doi.org/10.7554/eLife.49452.001

## Introduction

Adaptive phenotypic plasticity across a range of biological traits, from individual cell form and function to advanced learning, enables individuals to cope with fluctuating environments (*Pigliucci, 2001*; *Miner et al., 2005*). This flexibility is well-recognised in the diploid life stage, where a complex multicellular organism can generate adaptive plasticity in behaviour, morphology and physiology. However, far less is known about what plastic adaptive responses are possible at the seemingly simple unicellular gamete stage, when environmental variation can be profound. We therefore assessed whether males and females can adaptively vary sperm and egg function through gametogenesis in anticipation of impending functional environment.

Although sperm and eggs have universal primary roles that are vital for reproductive success, how they achieve these and the environments in which they must succeed can fluctuate considerably, both biotically and abiotically. Sperm almost always operate after ejaculation and release in a non-self and demanding environment, either within the female reproductive tract or through external fertilisation, and many factors that directly influence sperm function can vary profoundly across these environments (*Pitnick et al., 2009*; *Reinhardt et al., 2015*). Ova face a similarly challenging set of biotic and abiotic variables, especially in the many species that are oviparous where zygotes and embryos develop in eggs to hatch in the natural environment outside the mother (*Hinton, 1981*; *Gilbert, 2010*). These intrinsically variable environments for gamete function could lead to selection for adaptive plasticity, allowing males and females to improve their reproductive fitness by matching sperm and egg phenotypes through development in anticipation of different fertilisation and embryogenesis environments.

**\*For correspondence:**
M.Gage@uea.ac.uk

**Competing interests:** The authors declare that no competing interests exist.

One of the most important abiotic environmental variables is temperature (*Cossins and Bowler, 1987*; *Angilletta Jr., 2009*), especially in the context of climate change when thermal environments are expected to both warm and become much variable and extreme (*Perkins et al., 2012*; *Raftery et al., 2017*). Thermal variation has profound impacts on living systems, and numerous examples of adaptive plastic responses to temperature variability have been described, from acclimated mitochondrial function (*Pichaud et al., 2010*) and sex determination (*Warner and Shine, 2008*), up to complex shifts in behaviour (*Pateman et al., 2012*) and phenology (*Walther et al., 2002*). Temperature also influences gamete function across a number of levels, with sperm production and function being especially sensitive to warming (*Setchell, 1998*; *Sales et al., 2018*) and egg development being directly influenced by thermal regime (*Gillooly et al., 2001*; *Gillooly et al., 2002*). Here, we therefore use a combination of experimental approaches (overview on *Figure 1*) to examine whether males and females adaptively vary sperm and egg biology through gametogenesis at different temperatures, in anticipation of varying thermal environments for fertilisation and reproduction. We test for this potential in the flour beetle *Tribolium castaneum* which, like most sexually reproducing animals, is both ectothermic and oviparous (*Sokoloff, 1974*), and in which spermatozoal sensitivity to temperature is known (*Sales et al., 2018*). After exposing adult males and females and their gametes to different temperatures, we compare sperm and egg development and reproductive function within thermal regimes that mimic the increasingly variable conditions faced by

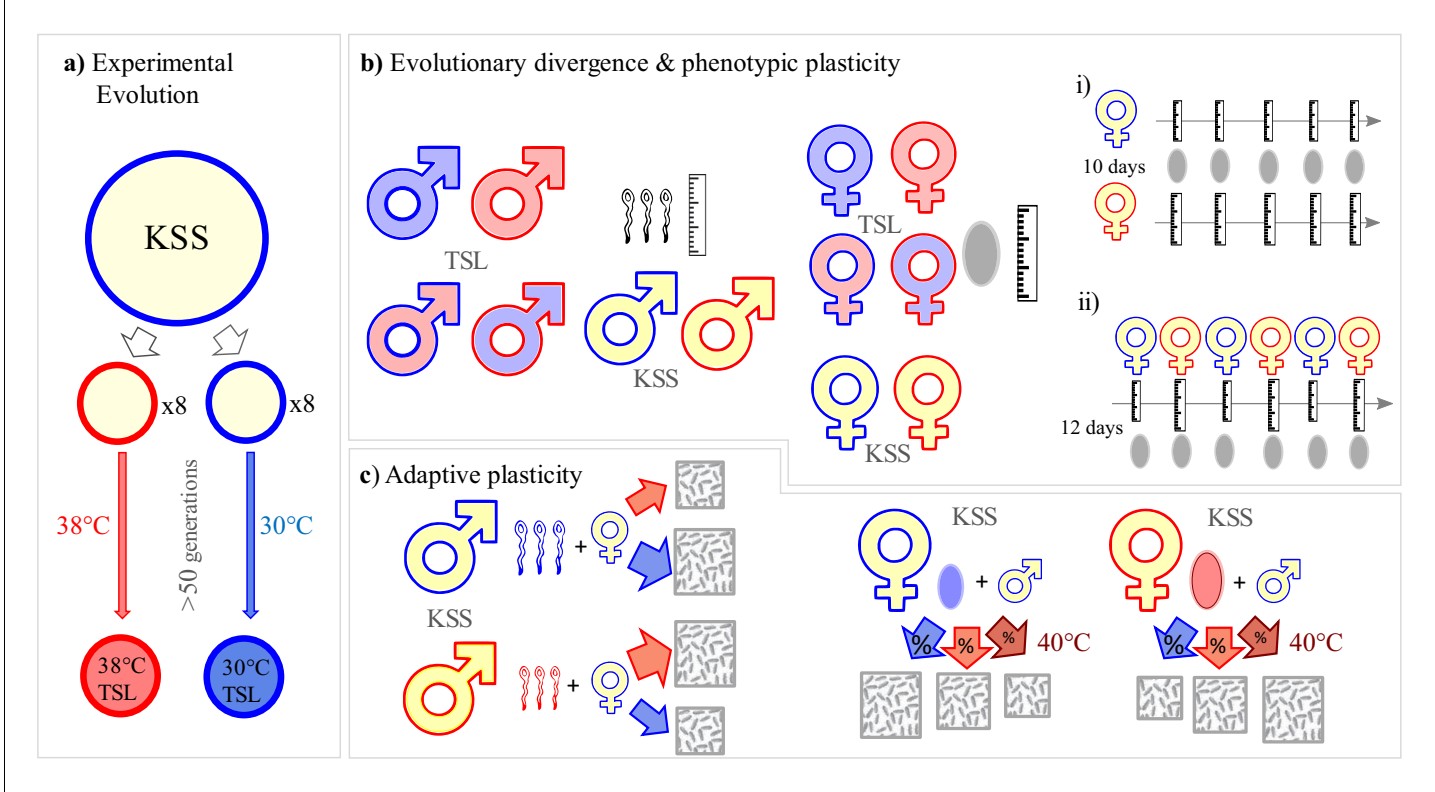

**Figure 1.** Experimental design. Overview of experiments investigating thermal adaptation and plasticity in sperm and egg biology and its adaptive significance. (a) Set up of long-term thermal selection lines (TSLs) maintained at 30˚C or 38˚C for 50+ non-overlapping generations with eight replicate populations per regime. (b) Experimental design to investigate gamete size divergence in 30˚C and 38˚C TSLs at their long-term evolving environments, short-term plasticity in gamete size measured after a single-generation of novel temperature exposures, and short-term plasticity in individuals from the ancestral KSS (Krakow Super Strain) population. Symbol fill colour represents long-term background (30˚C TSLs in blue, 38˚C TSLs in red and KSS in yellow) while outline colour represents short-term exposure temperature (30˚C blue, 38˚C red). bi) and bii) illustrate experiments on temporal patterns of short-term plasticity in egg size (see main text). (c) Experiments investigating the adaptive significance of sperm and egg morphological plasticity in KSS adults. Gametic divergence was achieved by having adults produce gametes at either 30˚C or 38˚C, whose performance was then tested at 30˚C, 38˚C or 40˚C.

DOI: https://doi.org/10.7554/eLife.49452.002

warmer tropical regions (*Perkins et al., 2012*). We find that thermal regime regulates gamete size development: in the short-term, as the developmental environment warms, eggs get larger and sperm become smaller. Having established these opposing male-versus-female responses, we use reciprocal transplant experiments to measure whether adaptive gamete plasticity exists. We find that warmer reproductive environments present challenges for reproduction. However, gamete plasticity enabled significant improvements in reproductive performance. Our findings reveal a potentially important route by which ectothermic populations can buffer their reproductive output against increasingly stressful and unpredictable temperature fluctuations under climate change.

## Results

### Sperm morphological divergence and plasticity

Following 54 generations of experimental evolution under increased temperature, we found that at both development temperatures sperm length differed by an average of ~4% between long-term selection regimes, with males from lines evolved at 38°C producing significantly longer sperm than males from lines evolved at 30°C (effect size $\beta = 3.4$, $t_{1,14} = 3.4$, p=0.004; eight lines per regime, five males per line, and five sperm per male, *Figure 2* and *Table 1*). In contrast, a within-generation increase in temperature during pupation and post-eclosion maturation reduced sperm length development irrespective of evolutionary background, with sperm produced at 38°C being ~7% shorter, and indicating developmental plasticity in sperm size ($\beta = -6.3$, $t_{1,24} = -11.2$, p<0.001; *Figure 2*). The interaction between selection regime and the development environment was not significant ($\beta = 0.5$, $t_{1,24} = 0.4$, p=0.672). Male body sizes of 30°C and 38°C thermal selection lines (TSLs) did

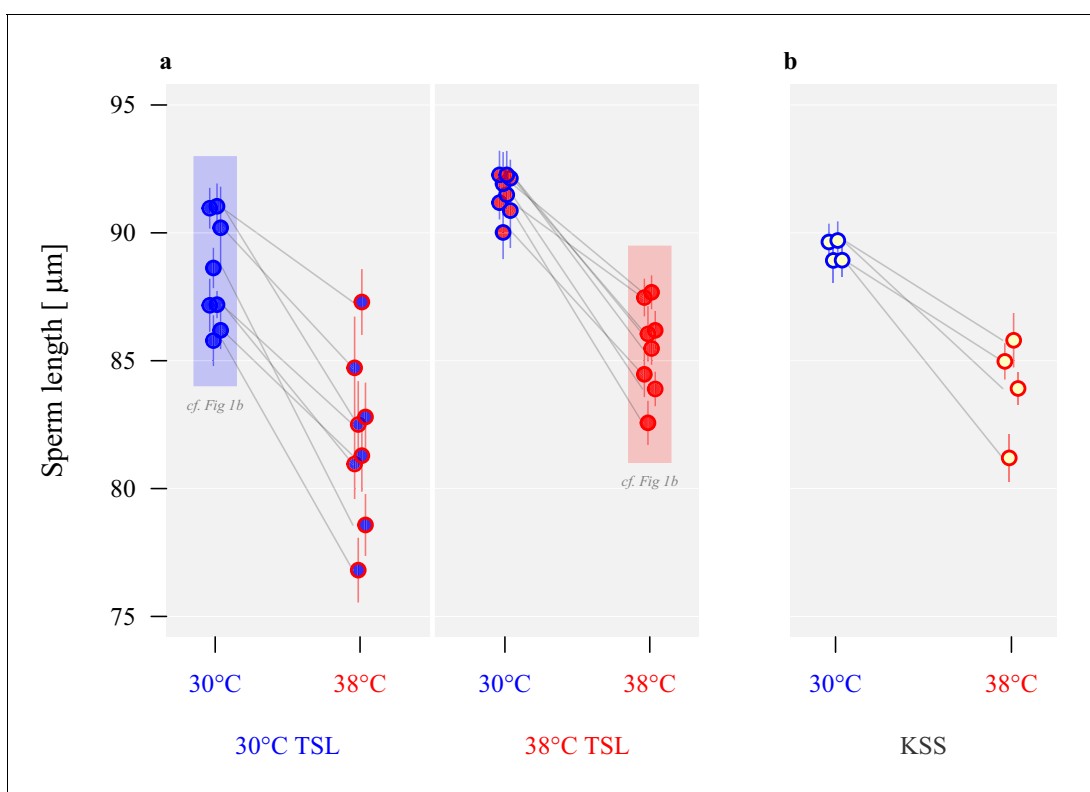

**Figure 2.** Long-term divergence and short-term plasticity in sperm size. Symbol fill colour represents long-term background (30°C TSLs in blue, 38°C TSLs in red and KSS in yellow) while outline colour represents short-term exposure temperature (30°C blue, 38°C red; *Figure 1b*). (a) Sperm length in μm ± SEM of the experimentally evolved TSLs, measured either following production from pupation in their long-term thermal environment (highlighted in shaded boxes) or at the reciprocal temperature. (b) Sperm length of mature KSS males exposed to either 30°C or 38°C from pupation through eclosion.
DOI: https://doi.org/10.7554/eLife.49452.003

**Table 1.** Sperm and egg length in relation to long-term selection and short-term exposure.

| Line | Temperature | Sperm length | N | Egg length | N |
|---|---|---|---|---|---|
| 30℃ TSL | 30℃ | 88.4 ± 5.2 | 40 | 615.4 ± 51.5 | 240 |
| 30℃ TSL | 38℃ | 81.9 ± 7.9 | 40 | 685.5 ± 42.2 | 240 |
| 38℃ TSL | 30℃ | 91.5 ± 4.9 | 40 | 638.4 ± 48.3 | 240 |
| 38℃ TSL | 38℃ | 85.5 ± 4.3 | 40 | 682.7 ± 42.7 | 240 |
| KSS | 30℃ | 89.3 ± 4.1 | 26 | 662.6 ± 39.4 | 180 |
| KSS | 38℃ | 84.3 ± 4.8 | 26 | 697.9 ± 48.4 | 180 |

Shown are mean, standard deviation and sample size (sperm: number of males; eggs: number of eggs) for sperm length and egg length measured in individuals from temperature selection lines (TSL) and the ancestral population (KSS), exposed to different temperatures from pupation onward (see main text and **Figures 1–3**).

DOI: https://doi.org/10.7554/eLife.49452.004

not differ ($F_{1,78}$ = 0.38, p=0.53), and similarly we found no evidence for body size divergence between 30℃ or 38℃ thermal environments from the pupal stage in the ancestral stock population (the Krakow Super Strain, KSS) ($F_{1,46}$ = 0.90, p=0.35).

## Egg morphological divergence and plasticity

Egg size also showed divergence and plasticity in relation to thermal regime. In contrast to sperm, egg size showed an increase in response to a hotter proximate temperature ($\beta$ = 70.0, $t_{1,14}$ = 13.9, p<0.001; **Figure 3a** and **Table 1**). However, there was a significant interaction between long-term selection regime and short-term temperature exposure ($\beta$ = −25.7, $t_{1,14}$ = -3.6, p=0.003; **Figure 3a**). Following 58 generations of experimental evolution (logistic contraints prevented simultaneous measurement at generation 54 when sperm lengths were assayed), eggs of females from TSLs evolved at 38℃ were larger than those of 30℃ TSL females when produced at 30℃ ($\beta$ = 23.0, $t_{1,14}$ = 3.2, p=0.006), but were very similar in size when produced at 38℃ ($\beta$ = −2.8, $t_{1,14}$ = -0.4, p=0.677).

Egg size was also thermally plastic in standard ancestral stock Krakow Super Strain (KSS) females, and showed significant divergence according to adult rearing and oviposition temperature when assessed in three experimental repeats. KSS females produced larger eggs when ovipositing at 38℃ compared to 30℃ ($\beta$ = 35.3, $t_{1,356}$ = 8.2, p<0.001; 50 females per group and 60 eggs measured per group at either rearing temperature, **Figure 3b**). In addition, this plasticity was shown by individual females ovipositing alone at either 30℃ or 38℃ ($\beta$ = 81.5, $t_{1,42}$ = 11.6, p<0.001; **Figure 3c**), and was reversible when females were alternated between 30℃ and 38℃ thermal environments ($\beta$ = 58.9, $t_{1,4}$ = 7.3, p=0.002; **Figure 3d**). Egg width was not different between the oviposition temperatures ($F_{1,198}$ = 0.1, p=0.686, **Figure 3—figure supplements 1** and **2**) and the interaction between oviposition temperature and egg length was not statistically significant ($t$ = −0.77, p=0.44), demonstrating that oviposition temperature increased egg volumes (see **Figure 3—figure supplement 2** for volume calculations).

## Adaptive sperm plasticity

As for the selection lines, sperm size was thermally plastic in ancestral stock KSS males, and showed significant divergence according to rearing temperatures (two experimental repeats), with KSS males producing significantly shorter sperm when reared at 38℃ compared to 30℃ ($\beta$ = −5.1, $t_{1,49}$ = -7.0, p<0.001; **Figure 2b** and **Table 1**). To test for the adaptive value of any functional plasticity, the performance of sperm from males of the same ancestral KSS population reared from pupation at either 30℃ or 38℃ was tracked by comparing the total number of offspring sired across 100 days of oviposition by KSS females at either 30℃ or 38℃ (**Figure 4a** and **Table 2**), by which time all females in the experiment had exhausted their viable sperm stores (**Michalczyk et al., 2010**) and ceased to produce offspring (**Figure 4b and c**). Reproduction was generally sensitive to the proximal thermal environment, with on average 299 (± 20.8, mean ± s.e.m.) offspring eclosing at 30℃, compared with 135 (± 10.3, mean ± s.e.m.) at 38℃ ($\beta$ = −5.8, $F_{1,104}$ = 69.8, p<0.001). Despite these overall differences, it was clear that the thermal regime in which sperm production and insemination took place also gave individual males a reproductive advantage when matched to the same thermal environment for

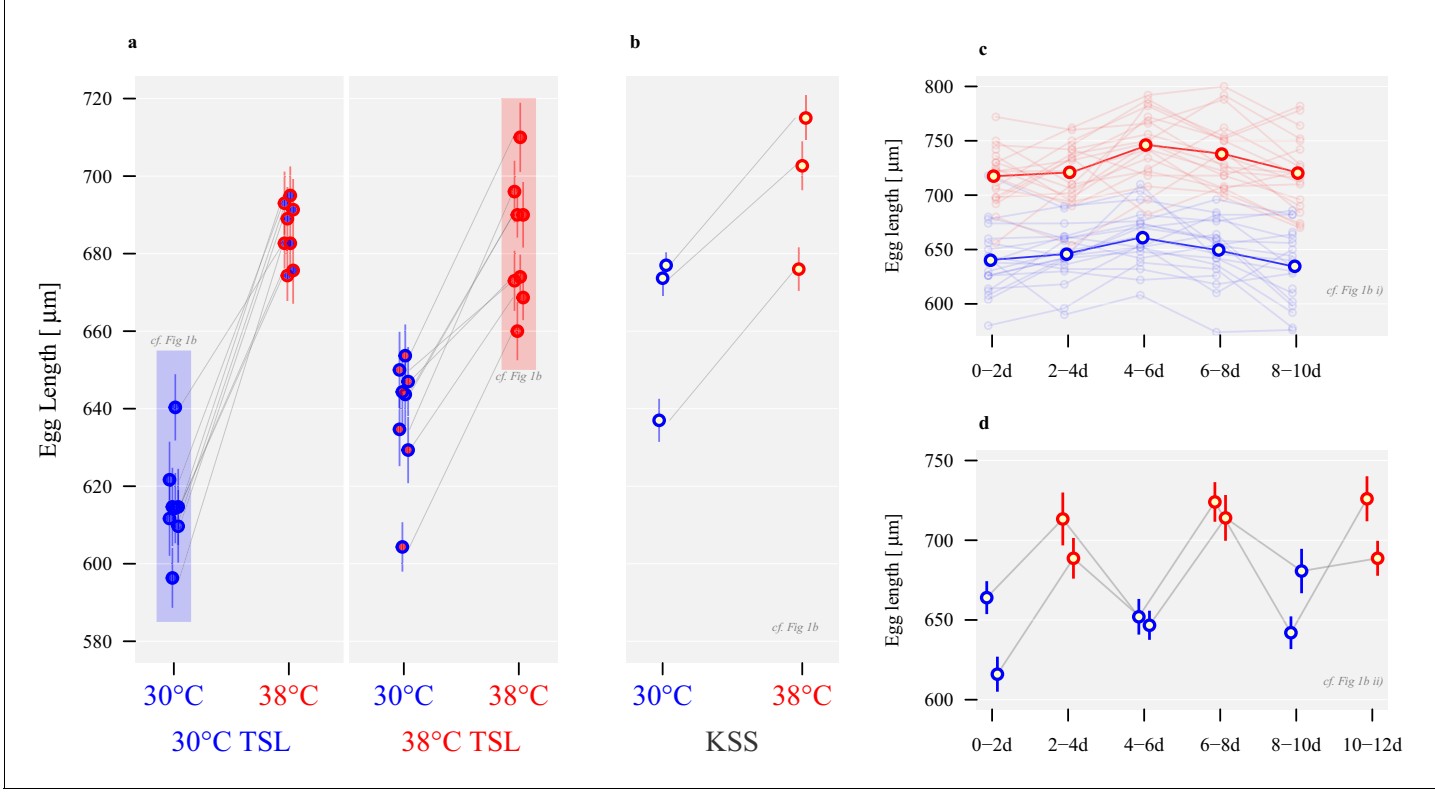

**Figure 3.** Long-term divergence and short-term plasticity in egg size. Symbol fill colour represents long-term background (30℃ TSLs in blue, 38℃ TSLs in red and KSS in yellow) while outline colour represents short-term exposure temperature (30℃ blue, 38℃ red; **Figure 1b**). (a) Egg length in µm ± SEM of the experimentally evolved TSLs, measured either following production in their long-term thermal environment (highlighted in shaded boxes), or at the opposite temperature. (b) Egg length produced by mature KSS females exposed to either 30℃ or 38℃ for mating and oviposition, measured across three experimental replicates (grey lines). (c) Egg length of KSS females mated to a standard KSS male ovipositing individually either at 30℃ or 38℃ for 10 days in two-day intervals. The two bold-face lines indicate the averages across all females within a given treatment, while thin lines show average values for individual females. (d) Egg length of groups of 50 females, mated to standard KSS males, and ovipositing alternately at 30℃ and 38℃ for 12 days in two-day intervals. Egg width did not differ between thermal regimes, demonstrating that oviposition temperature increased egg volumes (**Figure 3—figure supplements 1** and **2**).

DOI: https://doi.org/10.7554/eLife.49452.005

The following figure supplements are available for figure 3:

**Figure supplement 1.** Comparing differences in egg width (µm) of KSS females ovipositing at either 30℃ (blue) or 38℃ (red).

DOI: https://doi.org/10.7554/eLife.49452.006

**Figure supplement 2.** Scatter plot of egg morphology (egg length and width in µm) at the two ovipositing temperatures (blue squares 30℃; red circles 38℃).

DOI: https://doi.org/10.7554/eLife.49452.007

fertilisation and offspring development (**Figure 4a**). Indeed, the interaction between spermatogenesis temperature (sperm production) and offspring production temperature was highly significant ($\beta$ = 7.5, $F_{1,104}$ = 29.6, p<0.001), while the main effect of male treatment was not significant ($\beta$ = 0.7, $F_{1,104}$ = 1.1, p=0.290). Sperm from males exposed as pupae and maturing adults to 30℃ achieved greater reproductive success in the 30℃ reproductive environment than sperm from males that completed development at 38℃. By contrast, in the 38℃ reproductive environment sperm produced by males in the 38℃ treatment achieved double the reproductive success compared with sperm from males developed through the 30℃ treatment. Across 100 days of oviposition in the 30℃ reproductive environment, males reared at 30℃ sired an average of 349 (± 29.1, mean ± s.e.m.) offspring, while males reared at 38℃ sired 249 (± 27.1, mean ± s.e.m.) offspring. Using the same protocols in the 38℃ reproductive environment, males reared at 38℃ sired 180 (± 12, mean ± s.e.m. offspring), while males reared at 30℃ sired only 91 (± 12, mean ± s.e.m.) offspring (**Figure 4a** and

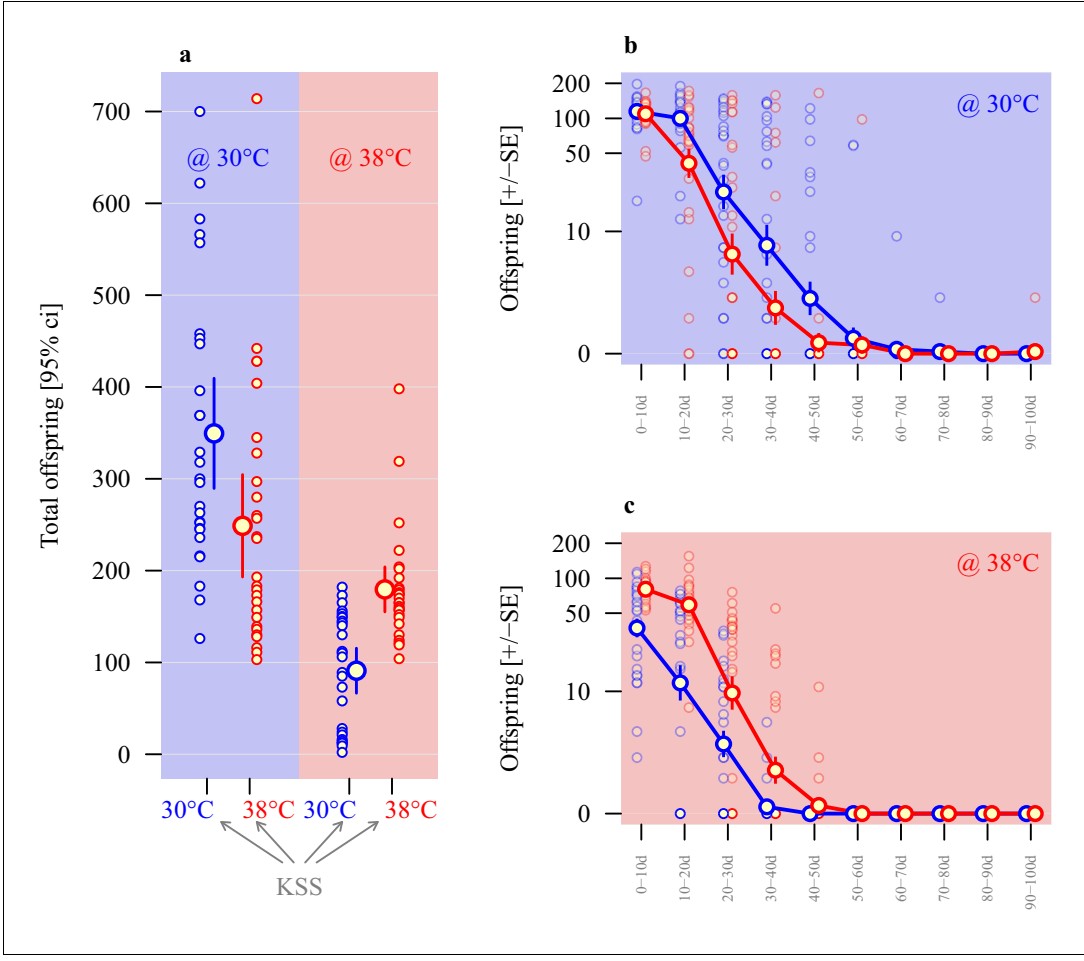

**Figure 4.** Adaptive thermal plasticity in sperm. Reproductive output of ancestral KSS males following a 24 hr mating bout with a single female. Symbol outline colour represents male short-term exposure temperature (30°C blue, 38°C red) while background colour indicates fertilisation and offspring development temperature (30°C blue, 38°C red; see *Figure 1c*). (a) Total offspring produced over a 100d period across ten 10 day blocks from sperm produced in either 30°C or 38°C conditions when functioning in either 30°C or 38°C reproductive environments. Temporal patterns in 30°C and 38°C environments are illustrated in (**b** and **c**), respectively (note the log-scale of the Y-axis). Analyses of individual male reproductive performance and average sperm length across a range of thermal regimes indicate a longer-sperm advantage in this system (*Figure 4—figure supplement 1*).
DOI: https://doi.org/10.7554/eLife.49452.008
The following figure supplement is available for figure 4:

**Figure supplement 1.** Sperm length (μm) and total reproductive output of KSS males used for the adaptive plasticity experiment (see *Figure 4a*).
DOI: https://doi.org/10.7554/eLife.49452.009

*Table 2*). Exploration of the temporal patterns of these effects showed a clear decline in reproduction across consecutive 10-day blocks, and that sperm matched to the fertilisation and development temperature consistently outperformed sperm that were thermally mismatched (*Figure 4b and c*). Our model comparison based on AIC values confirmed that the interaction between male rearing temperature and fertilisation temperature was important: the best model included the main effects of male temperature, offspring temperature and time, and the interaction between male and offspring temperature for both parts of the model, and additionally the interaction between rearing temperature and time for the zero-inflation model (*Supplementary files 1* and *2*).

**Table 2.** Adaptive thermal plasticity in sperm and eggs improves gamete performance.

| Line | Gamete production | Gamete performance | Sperm | | Eggs | |
|------|-------------------|--------------------|-------|---|------|---|
| | | | Offspring | N | Viability | N |
| KSS | 30°C | 30°C | 349.5 ± 151.5 | 27 | 90 ± 2% | 8 |
| KSS | 38°C | 30°C | 248.8 ± 140.9 | 27 | 80 ± 5% | 8 |
| KSS | 30°C | 38°C | 91.0 ± 61.8 | 27 | 78 ± 8% | 8 |
| KSS | 38°C | 38°C | 179.6 ± 61.8 | 27 | 80 ± 3% | 8 |
| KSS | 30°C | 40°C | | | 27 ± 5% | 8 |
| KSS | 38°C | 40°C | | | 36 ± 5% | 8 |

Sperm performance was measured by mating focal males to tester females and counting all offspring produced over a 100d period. Egg performance was measured as the proportion of eggs that developed into adult offspring, tested in groups of 50 eggs (see main text and **Figures 3** and **4**).
DOI: https://doi.org/10.7554/eLife.49452.010

## Adaptive egg plasticity

The performance of eggs produced by ancestral stock KSS females (mated to 30°C-reared KSS males) ovipositing at either 30°C or 38°C was compared by measuring egg-to-adult offspring viability of pre-counted groups of 50 eggs when incubated and reared at either 30°C, 38°C or 40°C (**Table 2**). There was a significant interaction between oviposition thermal regime and the environmental temperature treatments at which the eggs were incubated, and offspring hatched and developed ($z = -4.1$, p<0.001; **Figure 5**). Thus, there was similar evidence of adaptive egg plasticity as for sperm, although this was only evident in the 30°C and 40°C environmental treatments (**Figure 5** and **Table 2**). *Post hoc* testing showed that at 30°C eggs oviposited at 30°C achieved significantly greater egg-to-adult offspring viability rates than eggs oviposited at 38°C ($z = 4.6$, p<0.0001), whereas the opposite was true at 40°C ($z = 3.01$, p=0.01) where 30°C-oviposited eggs resulted in 25% fewer offspring than 38°C-oviposited eggs. In the 38°C environment, there was no evidence for any adaptive plasticity in egg biology, with the same relative number of eggs from the 30°C and 38°C regime females hatching and producing adult offspring (**Figure 5** and **Table 2**).

## Discussion

Our experiments revealed that: (1) gamete function and reproduction is highly sensitive to the local thermal environment; (2) developmental plasticity exists within sperm and eggs in response to temperature; (3) plastic responses in gamete size proceed in different directions in either sex; and (4) gamete plasticity is adaptive, enabling males and females to significantly improve their reproductive success via mechanisms that match sperm and egg development to the imminent thermal environment in which they must function.

Sperm production and function is known to be affected by many environmental variables (**Pitnick et al., 2009**; **Reinhardt et al., 2015**), with particular sensitivities to temperature (**Setchell, 1998**; **David et al., 2005**), which may be especially important in ectotherms where environmental temperature varies through the reproductive window (**Reinhardt et al., 2015**; **Walsh et al., 2004**). Such variation will directly influence important elements within the sperm storage and fertilisation environment (**Pitnick et al., 2009**; **Reinhardt et al., 2015**), including biophysical properties of fluids and membranes (**Kupriyanova and Havenhand, 2005**), mitochondrial metabolic sensitivity (**Sokolova, 2018**), flagellar function (**Humphries, 2013**) and haplotype integrity (**Paul et al., 2008**). Our transplant experiments reveal obvious challenges for gamete performance in warmer environments, with an overall halving of reproductive output when sperm and eggs are challenged to function at 38°C versus 30°C (**Figures 4** and **5**). Interestingly, between-line variance in sperm length also increased when males were forced to develop sperm at 38°C (**Figure 2a and b**), possibly indicating a response to thermal stress at the higher temperature.

Although we found evidence that laboratory selection across 54 generations at 38°C caused the overall evolution of ~4% longer sperm compared with the thermal line selection at 30°C, our within-generation experiments revealed consistent ~7% reductions in sperm length for all populations when produced through the warmer 38°C regime compared with 30°C (**Figure 2**). The different

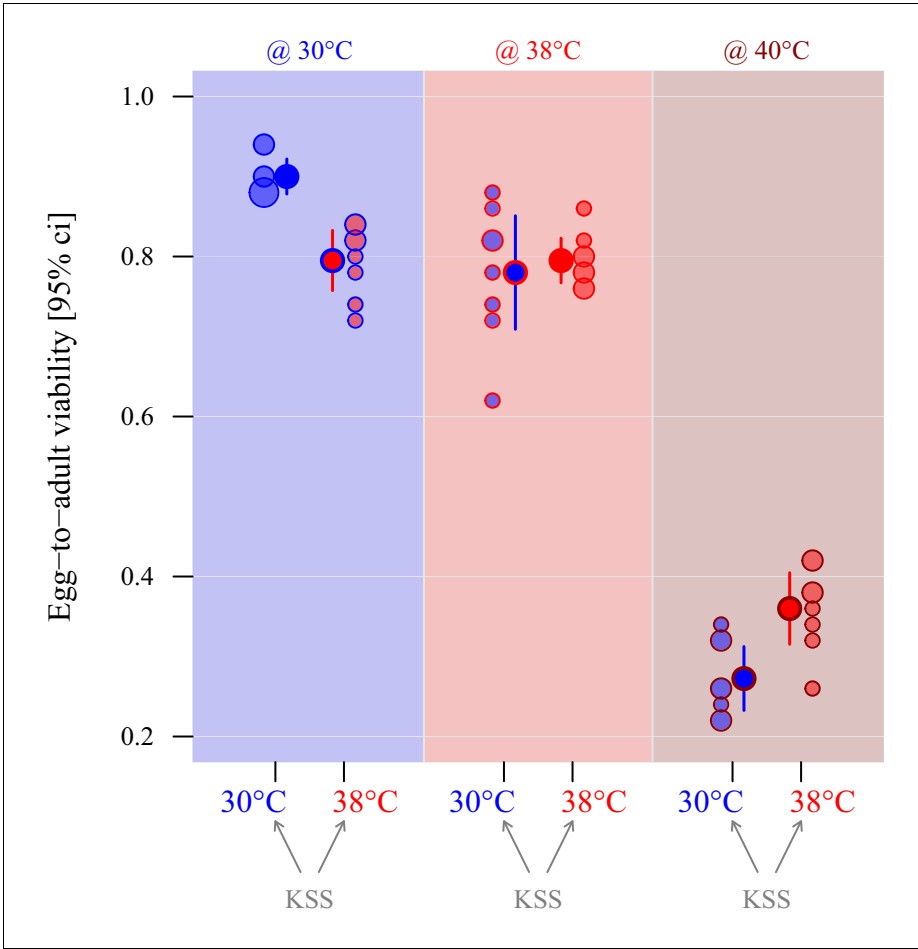

**Figure 5.** Adaptive thermal plasticity in eggs. Symbol fill colour represents production and oviposition temperature (30°C blue, 38°C red) for ancestral KSS females while outline and background colour indicate egg incubation, offspring hatching and development temperature (30°C blue, 38°C red, 40°C dark red; see *Figure 1c*). Egg-to-adult viability was measured in a fodder medium with 0% yeast, with eight replicate groups of 50 eggs per treatment combination. Point surface area is proportional to the number of observations with identical outcomes.
DOI: https://doi.org/10.7554/eLife.49452.011

sperm length responses to experimental evolution versus within-generation plasticity under different temperatures is puzzling, and may have arisen through indirect selection acting on the thermal lines for ~five years, given that the 38°C thermal selection lines showed generally longer sperm at both 30°C and 38°C rearing temperatures (*Figure 2a*). Recent work examining sperm length evolution in *T. castaneum* revealed that a history of heightened sexual selection led to the evolution of longer sperm (*Godwin et al., 2017*), so one possibility is that increased metabolic and developmental rates in the 38°C regime had elevated mating activity, and therefore promoted male-male competition and selection for increased sperm length. However, our current results (*Figure 4*) and past findings (*Sales et al., 2018*) indicate that warmer conditions generally reduce male reproductive fitness, and we find no evidence that operational sex ratios have deviated between our 30°C or 38°C selection line regimes, with both showing an average adult sex ratio of 50% (54 adults randomly sampled from six lines per regime: 30°C regime male ratios = 0.52, 0.46, 0.50, 0.52, 0.48, 0.52; 38°C regime male ratios = 0.50, 0.59, 0.48, 0.48, 0.52, 0.43); offspring sex ratios are also not changed following male exposure to 42°C heatwaves (*Sales, 2019*). Another possible explanation is that increased developmental rate at 38°C has hastened sperm ageing, leading to correlated changes in length. However, *T. castaneum* is a relatively long-lived insect, with adult males showing no change in fertility even after one year of lifespan (*Godwin, 2016*), so reproductive ageing differences over the few

days of the sperm length assays conducted here are unlikely to explain the overall sperm length increases in the thermal selection lines at 38°C. Another developmental possibility is that elevated spermatogenic rates at 38°C leads to the production of smaller cells, and longer-term selection has compensated for this reduction in sperm cell size within warmer regimes to evolve longer sperm. There is some evidence for longer-sperm advantage in *T. castaneum* (*Godwin et al., 2017*), and our results for individual male fertilisation success across a range of thermal regimes suggest that males producing longer sperm have improved reproductive fitness (*Figure 4—figure supplement 1*). Additional possibilities for indirect effects acting on sperm length in the 38°C selection regime include genetic bottlenecking at the start of selection if fertility and offspring production had been compromised, or heightened metabolic rate throughout the entire life cycle allowing improved access to key nutrients which may enable the development of longer sperm (*Godwin et al., 2017*).

Whatever the cause behind long-term experimental evolution of sperm length within our thermal lines, it is clear that short-term within-generation impacts of temperature have strong and direct effects on sperm development, with an experimental switch to warmer regimes for either the thermal selection lines or the ancestral stock population resulting in consistently clear reductions in sperm length (*Figure 2a and b*). Temperature variation during gametogenesis is known to affect sperm size in ectotherms (*Blanckenhorn and Hellriegel, 2002*; *Rohmer et al., 2004*), and cool and warm thermal extremes can reduce fertile function in tandem with reduced sperm length development (*Vasudeva et al., 2014*). However, no previous study has revealed that the thermal regime during spermatogenesis can shape sperm function to be optimal for the forthcoming thermal reproductive environment. We demonstrate clear evidence for adaptive plasticity in sperm function in anticipation of thermal regime, enabling males (and their mates) to increase their reproductive success by 40% to 100% when males are developmentally 'matched' to the temperature of the subsequent reproductive environment (*Figure 4*).

Adaptive plasticity in the production of sperm numbers is known in relation to environmental risks of male-male competition (*Wedell et al., 2002*), with the capability for spermatogenesis to increase in response to elevated risks of sperm competition (*Ramm and Stockley, 2009*; *Giannakara et al., 2016*). This male plasticity can also change individual sperm cell form and function: domestic cockerels (*Gallus gallus domesticus*) rapidly changed their sperm mobility in relation to their own competitive status (*Pizzari et al., 2007*), and male Gouldian finches (*Erythrura gouldiae*) adjusted sperm morphometry in relation to social factors (*Immler et al., 2010*). Similar changes occured in sperm velocity and density in Arctic charr (*Salvelinus alpinus*) when dominance switched (*Rudolfsen et al., 2006*), and broadcast-spawning ascidians (*Styela plicata*) produced larger, more motile sperm when adults were kept at densities with greater risks of sperm competition (*Crean and Marshall, 2008*). Far fewer studies have explored adaptive sperm plasticity in relation to abiotic variation, which is a gap because physico-chemical factors can greatly influence sperm function and also vary across reproductive environments (*Reinhardt et al., 2015*). Acclimation and in vitro fertilisation experiments with the broadcast spawning tubeworm *Hydroides diramphus* revealed adaptive plasticity in sperm and egg function in relation to salinity, with gametes performing best at salinities experienced by their parents prior to spawning (*Jensen et al., 2014*). Likewise, sticklebacks (*Gasterosteus aculeatus*) showed adaptive plasticity in spermatozoal sensitivity to the salinity and osmolarity signal for initiating flagellar motility (*Taugbøl et al., 2017*). To our knowledge, only two studies have investigated sperm plasticity in relation to temperature: male mosquitofish (*Gambusia holbrooki*) acclimated to cool (18°C) and warm (30°C) regimes for five weeks showed no signs of spermatozoal acclimation or change in thermal limits (*Adriaenssens et al., 2012*). Likewise, although warm conditions reduced sperm motility in brown trout (*Salmo trutta*), warm-acclimated males did not produce sperm with improved relative motility (*Fenkes et al., 2017*). Our transplant experiments revealed clear evidence for adaptive developmental plasticity in sperm function according to thermal regime, with males exposed to warmer 38°C temperatures producing sperm that enabled a doubling of reproductive success in warmer 38°C fertilisation and reproductive environments compared with sperm produced at 30°C (*Figure 4a and c*). The opposite also applied, with sperm produced in cooler 30°C regimes gaining ~40% greater reproductive success at 30°C compared with sperm produced at 38°C (*Figure 4a and b*). This plasticity will confer direct fitness advantages if the thermal regime through development before mating anticipates the temperature for sperm function and reproduction following insemination. Sperm manufacture proceeds rapidly in *T. castaneum*, with production of mature, functional gametes taking around four days at 30°C (*Fishman et al., 2017*). As in

many insects, fertilisation and oviposition proceeds within hours of mating in *T. castaneum* (*Fedina and Lewis, 2008*), so sperm production temperature will usually predict insemination, storage and fertilisation temperature.

In contrast to males, female *T. castaneum* produced larger gametes in the warmer environment (*Figure 3*), which is unusual for arthropods where smaller eggs are usually produced as temperatures increase (*Fox and Czesak, 2000*); but see *Seko and Nakasuji (2006)* and *Stillwell and Fox (2005)*. This egg size plasticity occurred within two days of exposure to the novel thermal regime, and was reversible (*Figure 3d*). Egg plasticity was also adaptive when comparing performance between the more extreme 30°C versus 40°C environments (*Figure 5*): 36% of eggs produced at 38°C generated viable offspring following development at 40°C compared with only 27% of eggs produced at 30°C. Conversely, 90% of eggs produced at 30°C generated offspring when developed at 30°C, versus 80% of eggs developed at 38°C (*Figure 5*). Adaptive plasticity in egg biology is known in relation to a number of environmental factors, but we believe this is the first study to demonstrate it under environmental warming. Adaptive egg plasticity in relation to biotic variation is shown by seed beetles (*Stator limbatus*), where females vary egg size in relation to anticipated host plant quality (*Fox et al., 1997*). Experiments show that females respond to a switch in host plant in a manner that maximises reproductive fitness by matching egg size (and number) to the host type (*Fox et al., 1997*; *Savalli and Fox, 2002*). Likewise, female cowpea weevils (*Callosobruchus maculatus*) are sensitive to levels of larval competition predicted by increased adult density, laying larger eggs to improve larval fitness (*Kawecki, 1995*). Broadcast-spawning ascidians (*S. plicata*) produce smaller eggs in high density populations, but their embryo-yielding ovicells are larger than eggs from low-density adults (*Crean and Marshall, 2008*). Similarly, female *Nasonia vitripennis* wasps adjust the sex ratio of their broods depending on whether they are first or second to parasitise a host, improving offspring fitness according to anticipated variation in local mate competition (*Werren, 1980*). Adaptive egg plasticity in relation to some abiotic variables is also recognised. Female stink bugs (*Podisus maculiventris*) detect reflectance at the site where they oviposit, and invest more protective pigment into eggs that will be exposed to stronger ultraviolet solar radiation (*Abram et al., 2015*). A number of studies have found variation in egg phenotypes according to temperature during development and oviposition (e.g. *Fox and Czesak, 2000*; *Ernsting and Isaaks, 2000*; *Blanckenhorn, 2000*), but few have identified that the changes are adaptive, with thermal variation creating physiological constraints or stress during egg production (*Fox and Czesak, 2000*). A notable exception is in *Bicyclus anynana* butterflies, where females lay larger (and fewer) eggs when they are exposed to lower temperatures during oviposition (*Fischer et al., 2003a*); reciprocal transfer experiments, as we employ here, show that this behaviour is adaptive (*Fischer et al., 2003b*). We identify adaptive plasticity in response to the upper ranges of reproductive tolerance by *T. castaneum* females, with adaptive matching through warmer egg development temperatures enabling a 33% improvement in offspring production in the hottest reproductive environment (*Table 2*, *Figure 5*).

Mechanisms for optimising gamete function in different thermal environments are to be uncovered, but four broad and related possibilities exist through: 1) optimising size, 2) physiological matching, 3) haploid selection, and/or 4) epigenetic modifications for offspring development. Although we observed opposing responses by sperm and egg sizes under temperature variation, it seems unlikely that this phenotypic variation is solely responsible for enabling improvements in reproductive success. Sperm length decreased as temperatures increased, but additional correlative analyses gave no indication that reduced sperm size per se improved reproductive performance in hotter thermal regimes, and the reverse was more evident (*Figure 4—figure supplement 1*), consistent with previous work showing that sperm elongation is costly and positively selected by competition (*Godwin et al., 2017*). It therefore seems more likely that changes in sperm physiology, rather than morphometry, allow plasticity in sperm function to match thermal reproductive environments. Thermal adaptations influencing cell physiology and biochemistry are known to exist within mitochondrial metabolism (*Egginton and Sidell, 1989*; *Guderley and St-Pierre, 2002*), essential for sperm function (*Ramalho-Santos et al., 2009*), and membrane properties influencing cell structure and physiology (*Cossins and Prosser, 1978*) for sperm and flagellum function (*Cardullo and Wolf, 1990*). In addition, Heat Shock Proteins, expressed throughout spermatogenesis in the testis, play key roles in sperm development (*Dun et al., 2012*). These adaptations for different thermal and hydrodynamic environments could be adaptively varied through spermiogenesis so that sperm function is matched to challenges facing sperm migration, storage and fertilisation in warmer

environments. Within-ejaculate haplotype selection provides another mechanism for improving off-spring performance (*Alavioon et al., 2017*).

Adaptive thermal plasticity may be possible via changes in egg size, because larger-volume eggs produced at higher temperatures will contain more fluids, and possibly a greater nutrient load, both of which could improve offspring viability under heat stress and desiccation (*Fischer et al., 2006*). However, within the 30°C environment, larger eggs produced at 38°C were outperformed by smaller eggs produced at 30°C, suggesting additional mechanisms beyond size benefits. As for sperm, thermally plastic traits that are essential for egg fertilisation and subsequent function could include differential mitochondrial activity (*Dumollard et al., 2007*) and egg plasma membrane properties (*Stein et al., 2004*). Heat Shock Proteins could also play vital roles in protecting egg development from thermal stress; recent work demonstrates that increased loading of HSP23 into *D. melanogaster* eggs improves embryo thermal tolerance (*Lockwood et al., 2017*).

In addition to the potential for adaptive plasticity within gamete function, reproductive fitness could be improved if short-term epigenetic modification through gamete development can pass adaptive information to the zygote, embryo and offspring (*Gannon et al., 2014*). The potential for adaptive transgenerational plasticity via the matriline and through transcription factors within eggs is increasingly recognised (*Ho and Burggren, 2010*), potentially enabling rapid responses to climate change via adaptive plasticity (*Diamond, 2018*). There is growing evidence that sperm also have the potential to be transcriptionally labile (*Immler, 2018*), passing environmentally-driven epigenetic information to offspring through histone or protamine modifications, haplotype DNA methylation remodelling, and/or small RNAs (*Donkin and Barrès, 2018*). Sperm carry complex payloads of coding and non-coding small RNAs which can be transcribed into the oocyte and embryo (*Dadoune, 2009*; *Hosken and Hodgson, 2014*) with conserved functions across mammalian and insect models (*Fischer et al., 2012*); the potential importance of these RNA transcripts remains largely unexplored (*Carrell, 2008*; *Carone et al., 2010*; *Sharma et al., 2016*). However, the possibility for transgenerational information transmission via sperm or egg epigenomes in response to environmental variation during gametogenesis is an obvious mechanism to enable adaptive thermal plasticity for populations facing the challenge of reproducing under climate change where increases in both thermal averages, maxima and variation are expected (*Perkins et al., 2012*; *Raftery et al., 2017*).

# Materials and methods

## Key resources table

| Reagent type (species) or resource | Designation | Source or reference | Identifiers | Additional information |
|---|---|---|---|---|
| Strain, strain background (*Tribolium castaneum*) | Krakow Super Strain ancestral stock and Thermal Selection Lines at 30°C and 38°C | KSS: *Dickinson, 2018 Sales, 2019* TSL: this paper and *Dickinson, 2018 Sales, 2019* | KSS & TSL30 or TSL38 | Live beetles |
| Biological sample (*Tribolium castaneum*) | Spermatozoa and ova | This paper | Sperm and eggs | Sperm recovered from sacrificed live males, eggs recovered from oviposition food medium |
| Software, algorithm | R Studio | R Studio (*RStudio Team, 2016*) in R (*R Development Core Team, 2017*) (version 3.4.1) | R Studio version 1.1.463 and R version 3.4.1 | |

## Model system, selection lines and thermal exposure

*Tribolium castaneum* flour beetles were maintained as previously described (e.g. *Lumley et al., 2015*; *Godwin et al., 2017*) in a 16 hr: 8 hr light: dark photoperiod at 60 ± 10% relative humidity in small stock populations in *ad libitum* fodder comprising 90% organic white flour, 10% brewer's yeast and a sprinkling of rolled oats to aid traction. Approval for the research was granted by UEA's Animal Welfare and Ethical Review Board. To identify males from females in mating pairs, individuals were marked with a small dot on the dorsal thorax with correction fluid (Tippex, France) (*Walker and Wineriter, 1981*). We used beetles from the Krakow Super Strain (KSS) in this study, a wild-type outbred lab strain created in 2008 by breeding together eleven different strains, and maintained since at 30°C (see *Dickinson, 2018*).

The thermal selection lines were established in 2010 and maintained since as eight independent replicates for either regime with synchronous generations at either 30 ± 1°C or 38 ± 1°C. Each line was reproduced through every adult generation using 100 haphazardly-selected, sexually mature adults (>ten days since pupal emergence). The mixed adults were free to mate and oviposit in 150 ml of fodder in 300 ml jars with mesh lids for one week, after which they were removed and the resulting eggs and larvae allowed to develop to the next generation at their respective temperatures. After 50+ generations of experimental evolution, gametes from males and females were assayed from eight independent lines in either thermal regime.

To measure the impacts of thermal regime on gamete biology, beetles were exposed to either 30°C, 38°C or 40°C temperatures in a controlled environment facility. Thermal exposure was regulated within a controlled environment facility held at 30°C, or in A.B. Newlife 75 Mk4 forced air egg incubators (A.B. Incubators, Suffolk, UK) at 38°C within the facility. 30°C is our standard rearing temperature, and the optimum laboratory temperature for population productivity in *T. castaneum* is 35°C (*Sokoloff, 1974*; *Sales, 2019*). 38°C approaches the upper thermal limit across more sensitive juvenile stages (*Dickinson, 2018*; *Sales, 2019*), and this temperature has been recorded in 150 countries (*Mherrera, 2019*). Our experimental exposure to this thermal regime will have relevance to more extreme thermal conditions and heatwaves, predicted to increase in frequency, severity and duration under climate change (e.g. *Perkins et al., 2012*), and which we know specifically constrain reproduction and sperm function in this system (*Sales et al., 2018*).

## Gamete measurements

Measures of mature sperm were performed following microdissection (see *Godwin et al., 2017*) and eggs were sieved from fodder following oviposition (*Figure 6*). Flour beetle eggs are oblong in shape, and size was quantified by measuring length across the long axis of rotational symmetry of the egg. Initial measures of length and width revealed a positive correlation between the two measures across 100 individual eggs, so a single length measure was used to quantify egg size. Eggs were sieved from the fodder using mechanical sieves (pore size: 300 μm, Endecotts Ltd, London, UK), and placed on a dark tile using a fine paintbrush. Eggs from the fodder were coated with a single layer of flour, and were measured at x30 magnification using a Zeiss Discovery V12 binocular microscope, AxioCam MRc5 camera and AxioVision V5.1 imaging software. Mature sperm were recovered from the base of the testicular follicles dissected out of males frozen at –6°C. Follicles were isolated on microscope slides in 30 μl drops of insect saline (0.9% NaCl), and then ruptured using fine forceps under an Olympus SZX9 binocular microscope. Once ruptured, sperm were dispersed by spreading out the area of the saline drop across the microscope slide using fine forceps, and the smears left to dry at room temperature so that the sperm cells lay in flat two-dimensional planes adhering to the glass. Images of intact spermatozoa were captured at 600X magnification under dark-field phase contrast using an Olympus BX41 microscope connected to a GT Vision GX CAM digital camera and GXCapture 8.2 software (GT Vision Ltd, UK). Sperm length was measured by creating a segmented line that traced the entire length of the cell using the 'ImageJ' analysis package and segmented line tool (*Schneider et al., 2012*). Previous work shows this approach has high repeatability (*Godwin et al., 2017*).

## Long-term and short-term gamete divergence (*Figure 1a and b*)

We assessed the impact of temperature on development of gamete sizes following both long-term and short-term variation in thermal regime. Responses to long-term variation were measured

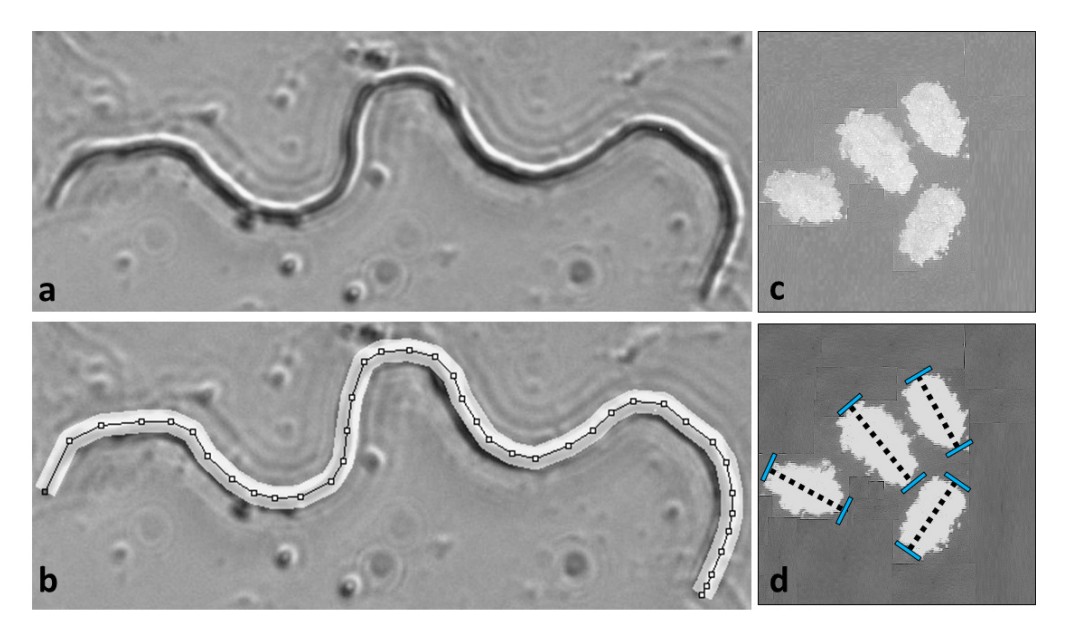

**Figure 6.** Gamete measurements were performed on mature sperm dissected from males and eggs following oviposition. Sperm length was measured at 600X magnification in Image J by drawing segmented lines along the backbone of the cell (**a** and **b**). Egg lengths were measured along the long axis of each ovoid at 30X magnification (**c** and **d**). Further details in *Godwin et al. (2017)*.

DOI: https://doi.org/10.7554/eLife.49452.012

following five years of selection (50+ generations) within replicate lines maintained in either 'warm' (30 ± 1˚C) or 'hot' (38 ± 1˚C) conditions. To measure impacts of short-term thermal variation we created duplicates of each of the eight replicate 30˚C and 38˚C lines at generation 54 to measure sperm effects and 58 for egg effects, and reared these at both 30˚C and 38˚C (*Figure 1b*) using a balanced, factorial design. Logistic contraints prevented simultaneous measurement at generation 54. To measure effects on sperm, males were exposed to either temperature from the pupal stage in 6 cm Petri dishes containing 15 ml of standard fodder, with adults allowed to emerge in groups of 20 per dish at their treatment temperature (*Figure 1b*). Ten days later, when reproductively mature, they were frozen for dissection and sperm measurement. We also measured full body size of males from both regimes ($N$ = 80; five males from each of eight lines across two regimes). Sperm length variation was measured in five males per replicate line, recovering sperm from frozen males and measuring the total length of five sperm per male (=200 sperm measures from 40 males across eight lines in each of four thermal x selection treatment combinations).

To measure temperature effects on eggs, mature adult females from eight replicate 30˚C and 38˚C lines at generation 58 were allocated to oviposit for two days at the two temperatures in a fully factorial design. Two days after the introduction of adults, we sieved oviposited eggs from the line's fodder, isolating them for measurement. Egg length measurement followed a balanced design across temperature regimes and lines, with 30 eggs measured from each replicate line at either temperature regime (=960 egg measures across eight replicate lines of either selection regime and two short-term temperature exposures).

We also measured gamete size plasticity in the ancestral KSS stock population maintained at 30 ± 1˚C, following methods for testing gamete size plasticity as above. Males were developed from pupae at either 30˚C or 38˚C and, ten days following emergence, five sperm and body length were measured from each of 26 males per treatment. Eggs were measured in three experimental blocks, within which two groups of 50 KSS females (eclosed and mated at 30˚C with standard KSS males also developed at 30˚C) oviposited at either 30˚C or 38˚C for two days in 100 ml jars with perforated lids and 80 ml of standard fodder. We measured 60 eggs per oviposition group (total $N$ = 3600 eggs). Two additional tests measured short-term thermal impacts on egg size: the first measured

egg size plasticity within individual females ($N$ = 20 at 30°C and $N$ = 20 at 38°C), with 40 KSS females mated to individual 30°C-developed KSS males for 24 hr at 30°C, after which 20 each were randomly allocated to oviposit at either 30°C or 38°C in 4 ml vials containing 0.5 g of standard fodder, transferring females to new vials every two days for a total of ten days, and eggs measured (length and width, μm) from vials immediately after females had been transferred on (*Figure 1bi*). The second test examined short-term reversibility of egg size in a group of 50 KSS mated females which were alternated for oviposition between 30°C and 38°C thermal regimes every two days for a total of 12 days, starting at 30°C (*Figure 1bii*). Females had been mated to standard 30°C-developed KSS males, and kept in 100 ml jars containing 80 ml of standard fodder, and 30 eggs per jar were measured immediately after transfers (total $N$ = 180 eggs).

## Assessing the adaptive significance of gamete plasticity (*Figure 1c*)

To measure adaptive plasticity in sperm, we reared KSS male pupae through either 30 ± 1°C or 38 ± 1°C temperatures as described above, and then tested the relative performance of sperm from eclosed males within KSS females and ova that were maintained in either 30 ± 1°C or 38 ± 1°C fertilisation and oviposition regimes for 100 days. Male pupae were isolated from the KSS stock population and completed development to eclosion in groups of 20 in 6 cm plastic Petri dishes with *ad libitum* fodder at either 30 ± 1°C or 38 ± 1°C. Three of these male groups ($N$ = 60 pupae) were reared and maintained at 30°C, and three ($N$ = 60 pupae) at 38°C.

When sexually mature at 10 days post eclosion, individual males from either the 30°C or 38°C eclosion regime were paired with similar-aged, marked virgin females from the KSS stock population (reared at 30°C) in 7 ml vials containing 0.5 g of fodder. Pairs were allowed to mate at the male's eclosion temperature. After 24 hr, pairs were separated, and females isolated in individual 6 cm Petri dishes containing 10 g of fodder. Half the mated females within either male thermal treatment group were allowed to oviposit at 30 ± 1°C, and the other half at 38 ± 1°C, with $N$ = 27 females in each of the four oviposition groups. Thus, we executed a fully factorial and balanced experiment in which females developed, fertilised and oviposited eggs at either 30 or 38°C, using sperm that had been produced at either 30°C or 38°C. Every ten days, females were transferred to new Petri dishes containing fresh fodder, for a maximum of 100 days (ten x 10 day blocks), by which time females had ceased to produce fertile eggs (following a single mating period, female *T. castaneum* typically use up all viable sperm within 100 days, after which a new mating allows resumption of fertility and offspring production [*Michalczyk et al., 2010*]). The number of adult offspring emerging from each Petri dish across up to 100 days of oviposition at either 30°C or 38°C quantified the reproductive success of each pair, comparing performance of sperm developed at either 30°C or 38°C when challenged with functioning at either 30°C or 38°C.

To measure the adaptive significance of egg plasticity, we isolated eggs that had been developed and laid at either 30 ± 1°C or 38 ± 1°C temperatures from KSS adults, and then tested their egg-to-adult viability through either 30 ± 1°C, 38 ± 1°C or 40 ± 1°C thermal regimes. To generate phenotypic divergence in egg biology, groups of 300 females previously mated to KSS males at 30°C oviposited for 2 days at either 30 ± 1°C or 38 ± 1°C (two groups at either temperature) in 1200 ml tubs containing 600 ml of standard fodder. 600 eggs per group were sieved and isolated from the flour, counted, and transferred in clutches of 50 to develop in 100 ml jars containing yeast-free fodder (applying stronger environmental selection on offspring development). Egg clutches produced at either 30°C or 38°C were transferred to hatch and develop at either 30°C, 38°C or 40°C, with eight groups assayed in each of these three temperature treatments (=2400 eggs assayed across a total of 48 treatment groups).

## Statistical analysis

All analyses were carried out using R Studio (*RStudio Team, 2016*) (v 1.1.463) in R (*R Development Core Team, 2017*) (version 3.4.1) with *Rmisc* (*Hope, 2013*), *multcomp* (*Hothorn et al., 2008*), *car* (*Fox and Weisberg, 2011*), *MASS* (*Venables and Ripley, 2002*), *glmmTMB* (*Brooks et al., 2017*) and *lmerTest* (*Kuznetsova et al., 2017*) packages for data exploration and analysis. Graphical figures were plotted using *ggplot2* (*Wickham, 2011*). Unless otherwise specified, all data were analysed using Linear Mixed Models (LMM) and Generalised Linear Mixed models (GLMM) in *lme4* (*Bates et al., 2015*), with the specific approach for each set of results described below. All data

generated from the experiments described above were included for analysis, and all replication is biological.

To determine the appropriate error distributions the relationship between the variance and the mean of the response variable and the assumptions for data distribution were checked (*Crawley, 2012*). Models were fitted using Restricted Maximum Likelihood (REML) methods to enable refinement and validation (*Thomas et al., 2013*). Residuals from linear models were checked for violations of the assumptions of normality and homoscedasticity. Significance of fixed effects in LMMs were obtained using t-tests with Satterthwaite's approximation for degrees of freedom implemented in *lmerTest* (*Kuznetsova et al., 2017*). To facilitate the interpretation of main effects in the presence of interactions, we centred the contrasts between factors with two levels by coding them as minus 0.5 and 0.5, respectively (*Schielzeth, 2010*).

*Total sperm length divergence* following long-term evolution and short-term temperature exposure in the thermal selection lines (TSL) was analysed using an LMM, with the selection regime (30℃ or 38℃), the exposure temperature and their interaction entered as fixed factors, and the replicate male (1–5) as a random factor, nested within each of the eight replicate lines. To account for our fully factorial design, we additionally included random slopes for our replicate lines. Divergence in KSS sperm length was analysed using an LMM with thermal environment (30℃ or 38℃) as a fixed effect, and male (five sperm from each of 26 males per thermal environment) nested within the two experimental runs as random effects.

*Adaptiveness of sperm length plasticity* was assessed using a General Linear Model (LM) on the total number of offspring per mating pair (after square root transformation), with male temperature treatment (30℃ or 38℃), fertilisation temperature (regime; 30℃ or 38℃) and their interaction as explanatory variables. To additionally explore temporal variation in offspring production, we additionally ran zero-inflated models with gaussian distribution on offspring counts, implemented in *glmmTMB* (*Brooks et al., 2017*). Almost all females ceased to produce offspring in the last few 10-day blocks which has previously been shown to be due to sperm limitation (*Michalczyk et al., 2010*). We ran a selection of models, and selected the model with the lowest AIC value as our best model (*Supplementary files 1* and *2*). In our conditional full model, we included KSS male treatment, fertilization temperature, time (using blocks as a continuous variable), and two- and three-way interactions as fixed effects. We included random intercepts for 10-day blocks and random slopes for individual pairs to account for repeated measures across time (*Schielzeth and Forstmeier, 2009*). Our zero-inflated full model included male treatment, fertilization temperature (regime), time, and two- and three-way interactions as fixed effects.

*Egg length divergence* in groups of TSL females was analysed analogous to sperm length, using an LMM with selection regime (30℃ or 38℃), short-term exposure temperature and their interaction as fixed effects, and replicate line (1–8) as a random factor, including random intercepts and slopes. Egg length plasticity in the KSS stock following oviposition at either 30℃ or 38℃ was analysed in an LMM with thermal regime as a fixed effect and experimental block included as a random effect. Plasticity in egg morphology in individual females was analysed in an LMM with thermal regime as a fixed effect and random intercepts for female ID as well as for two-day blocks. We additionally modelled temporal trends by including two-day blocks as a continuous fixed effect, and random slopes for individual KSS females, but found no evidence for temporal trends on egg morphology (p=0.9). To analyse reversibility of egg size plasticity we used an LMM with thermal regime as a fixed effect and two-day blocks as a random effect. Additionally, on a subset of the eggs, we measured egg width to quantify subsequent changes in morphology and overall volume with oviposition temperature (N = 20 KSS females per thermal exposure and five eggs per female measured). To assess the correlation between egg length and egg width (egg morphology) at the two oviposition temperatures, a simple LM was fitted to the data with egg width as a response variable, and egg length, oviposition temperature and their interaction as predictor variables.

*Adaptive developmental plasticity in egg function* was assessed by testing the reproductive performance of replicate groups of 50 eggs produced by females at either 30℃ or 38℃ when hatching and developing in 30℃, 38℃ or 40℃ thermal environments. Egg performance was analysed using a Generalised Linear Model (GLM) with binomial error structure (logit link) in which egg production temperature (30℃ or 38℃), developmental thermal environment (30℃, 38℃ or 40℃), and their interaction, were entered as fixed effects. We included the number of successes (developed offspring) and failures (eggs that failed to hatch/develop) using the *cbind* function, and confirmed that our

model was not overdispersed. Finally, temperature impacts on male body size were analysed using a simple LM. Female body sizes were not assessed because egg size plasticity tests under thermal variation were conducted on already-emerged mature adult females.

An overview of sample sizes is given in *Tables 1* and *2*. Box plots indicate the median and interquartile ranges (IQR), with whiskers indicating data within 1.5 IQR. A central filled marker indicates the mean of the sample.

## Data accessibility statement

All data generated and analysed in this study, together with R codes, are openly provided as an associated source file in our Dryad Data Repository with the identifier: https://doi.org/10.5061/dryad.83bg17q.

## Acknowledgements

This work was supported by NERC grant NE/K013041/1, a Commonwealth Rutherford Fellowship (to RV) by the Commonwealth Scholarship Commission, UK, and a Swiss National Science Foundation fellowship (to AS; grant number P300PA_177906). We thank Maya Krishnan Kumar for help with *ggplot2*, and the eLIFE editorial team and three expert referees for reviewing the work in detail and substantially improving the manuscript.

## Additional information

### Funding

| Funder | Grant reference number | Author |
| --- | --- | --- |
| Natural Environment Research Council | NE/K013041/1 | Matthew Gage |
| Commonwealth Scholarship Commission | Commonwealth Rutherford Fellowship | Ramakrishnan Vasudeva |
| Swiss National Science Foundation | P300PA_177906 | Andreas Sutter |

The funders had no role in study design, data collection and interpretation, or the decision to submit the work for publication.

### Author contributions

Ramakrishnan Vasudeva, Conceptualization, Data curation, Formal analysis, Supervision, Validation, Investigation, Visualization, Methodology, Writing—original draft, Project administration, Writing—review and editing; Andreas Sutter, Conceptualization, Data curation, Formal analysis, Supervision, Validation, Visualization, Writing—original draft, Project administration, Writing—review and editing; Kris Sales, Data curation, Formal analysis, Validation, Investigation, Visualization, Methodology, Writing—review and editing; Matthew E Dickinson, Data curation, Investigation, Methodology, Writing—review and editing; Alyson J Lumley, Data curation, Supervision, Investigation, Methodology, Project administration, Writing—review and editing; Matthew JG Gage, Conceptualization, Resources, Supervision, Funding acquisition, Validation, Investigation, Methodology, Writing—original draft, Project administration, Writing—review and editing

### Author ORCIDs

Ramakrishnan Vasudeva https://orcid.org/0000-0002-3831-0384
Andreas Sutter http://orcid.org/0000-0002-7764-3456
Matthew E Dickinson https://orcid.org/0000-0001-5023-4233
Matthew JG Gage https://orcid.org/0000-0003-3318-6879

### Ethics

Animal experimentation: This study was approved by, and followed strict guidelines to, the University of East Anglia's Animal Welfare and Ethical Review Board.

## Decision letter and Author response
Decision letter https://doi.org/10.7554/eLife.49452.019
Author response https://doi.org/10.7554/eLife.49452.020

## Additional files

### Supplementary files
• Supplementary file 1. Table model summary of the best *glmmTMB* model (lowest AIC value; see *Supplementary file 1* Table 2) for reproductive output of males reared at 30°C or 38°C (Treatment) with offspring developing at 30°C or 38°C (Regime).
DOI: https://doi.org/10.7554/eLife.49452.013

• Supplementary file 2. Table overview of models for adaptive significance of sperm plasticity. Models were run using *glmmTMB* and were sorted along ascending AIC values. All conditional models additionally included random intercepts for ten-day blocks as well as random intercepts and random slopes for individual males.
DOI: https://doi.org/10.7554/eLife.49452.014

• Transparent reporting form
DOI: https://doi.org/10.7554/eLife.49452.015

### Data availability
Data accessibility statement: All data generated and analysed in this study, together with R codes, are openly provided as an associated source file in our Dryad Data Repository with the identifier: https://doi.org/10.5061/dryad.83bg17q.

The following dataset was generated:

| Author(s) | Year | Dataset title | Dataset URL | Database and Identifier |
|---|---|---|---|---|
| Ramakrishnan Vasudeva, Andreas Sutter, Kris Sales, Matthew J. G. Gage | 2019 | Data from: Adaptive thermal plasticity enhances sperm and egg performance in a model insect | https://doi.org/10.5061/dryad.83bg17q | Dryad, 10.5061/dryad.83bg17q |

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
