## [Decision Letter]

Thank you for submitting your article "Adaptive thermal plasticity enhances sperm and egg performance in a model insect" for consideration by *eLife*. Your article has been reviewed by three peer reviewers, one of whom is a member of our Board of Reviewing Editors, and the evaluation has been overseen by Patricia Wittkopp as the Senior Editor. The following individuals involved in review of your submission have agreed to reveal their identity: Neil J Gemmell (Reviewer #2); Tom Pizzari (Reviewer #3).

The reviewers have discussed the reviews with one another and the Reviewing Editor has drafted this decision to help you prepare a revised submission.

Summary:

The authors make use of the Tribolium flour beetle model to assess the potential for thermal adaptation in sperm and egg function. Exposing flour beetles to increased temperature under both long-term experimental evolution and short term acclimation, the authors found that thermal regime regulates gamete size development in both males and females. They find that long-term selection regimes at high temperatures (38°C) led to males producing sperm 4% longer than those evolved at lower temperatures (30°C). However, short term changes in temperature generated the opposite pattern: sperm production in males subjected to 38°C within a single generation produced sperm 7% shorter than those exposed to 30°C. Females also showed a response to both the long and short term selection regimes, producing longer eggs at higher temperatures.

Using a series of well-conducted thermal translocation experiments, the authors show that this gametic phenotypic plasticity was adaptive. Both sperm and eggs produced in warmer conditions had superior reproductive performance in warmer environments, and vice versa for colder conditions and reproductive environments. Further, in warmer environments, the gamete plasticity observed enabled males to double their reproductive success, and females could increase offspring production by one-third. Collectively the authors' work identifies a new dimension in gametic plasticity important for understanding how organisms may respond to the increasing thermal variation in the natural environment.

This is an excellent study, presenting a uniquely exhaustive experimental investigation into the consequences of environmental temperature for gametic phenotype and related fitness consequences in an ectotherm invertebrate model species. While the biological ramifications of climatic change, including environmental temperature, have been the focus of intense and increasing research interest in recent years, surprisingly little remains known about the impact on gametic performance and associated implications for reproductive fitness. The study therefore makes an influential contribution to a topical and unresolved question.

Essential revisions:

Perhaps the most intriguing result is the strong contrast in phenotypic response in sperm length under long-term and short-term regimes. Under long term selection to higher temperatures sperm get longer, while under the short term regime, they got shorter. I find this a puzzling result, which I cannot easily reconcile. The authors have put forward some plausible explanations, which I think are sensible, but I did wonder if other factors might contribute to or explain this result. First, was there potential for the sex ratios in the selection lines to have altered as a consequence of thermal exposure, which may have affected the level of sperm competition across some treatments? I wondered if this might be a particular factor in explaining the difference in sperm length response between short-term and long-term selection experiments? I know the authors are expert in this realm, so assume they controlled for this possibility, but some details on whether sex ratios differed and how these were handled might be useful to include. Similarly, I wondered if there were any data on sperm volumes or concentrations. While sperm length clearly responded to the selection regime, I wondered if the authors might also know how sperm number, ejaculate volume and motility altered as there are often strong interactions among these traits? Probably these traits are not easily measured in flour beetles, but if there are data, they might further strengthen this already impressive piece of work. Last, I wondered if there might be an ageing effect that was driving some of the findings observed. In particular, I was curious as to whether males age faster and/or sperm mature faster at higher temperatures. Might part of the effect observed be driven by male age and ejaculate age? Perhaps there is a way to factor in degree days to control for any variation that might exist among experiments?

The authors need to justify the experimental temperatures the beetles were exposed to. In special, 38 C is not a yearly average temperature that exists anywhere in the world. Even the lower temperature (30 C) is not a common average temperature anywhere in the world. And very unlikely it will exist, even under global warming scenarios. So, the paper can be better framed in the context of seasonal temperature variability (growing seasons, summer, etc.) and not necessarily in a global warming context. Comparing only two temperatures 30 vs. 38 C is not ideal, besides being temperatures that are very far apart from each other, it is impossible to draw a general trend when you only have one comparison. It would have been great had the manuscript had an intermediate temperature treatment, at 34 C (or so).

---

## [Author Response]

Essential revisions:Perhaps the most intriguing result is the strong contrast in phenotypic response in sperm length under long-term and short-term regimes. Under long term selection to higher temperatures sperm get longer, while under the short term regime, they got shorter. I find this a puzzling result, which I cannot easily reconcile. The authors have put forward some plausible explanations, which I think are sensible, but I did wonder if other factors might contribute to or explain this result. First, was there potential for the sex ratios in the selection lines to have altered as a consequence of thermal exposure, which may have affected the level of sperm competition across some treatments? I wondered if this might be a particular factor in explaining the difference in sperm length response between short-term and long-term selection experiments? I know the authors are expert in this realm, so assume they controlled for this possibility, but some details on whether sex ratios differed and how these were handled might be useful to include. Similarly, I wondered if there were any data on sperm volumes or concentrations. While sperm length clearly responded to the selection regime, I wondered if the authors might also know how sperm number, ejaculate volume and motility altered as there are often strong interactions among these traits? Probably these traits are not easily measured in flour beetles, but if there are data, they might further strengthen this already impressive piece of work. Last, I wondered if there might be an ageing effect that was driving some of the findings observed. In particular, I was curious as to whether males age faster and/or sperm mature faster at higher temperatures. Might part of the effect observed be driven by male age and ejaculate age? Perhaps there is a way to factor in degree days to control for any variation that might exist among experiments?

We have added a major section to the Discussion where we offer additional explanations for the general evolution of longer sperm in the warmer regime, despite short-term responses leading to shorter sperm. We include the reviewer suggestions for factors arising from sexual selection and/or developmental ageing. We also suggest nutrient access through the whole life cycle for spermatogenesis as a potential factor, since nutrient limited males develop shorter sperm in T. castaneum (Godwin et al., 2017). Finally, we consider the possibility that longer-term evolution in a warmer 38^o^C environment, where sperm are pushed by short-term effects (perhaps via faster development rate) to be shorter, has led to compensatory selection for sperm elongation.

We do not have data on sperm number/volume/concentration or motility in our thermal lines, because achieving meaningful measures of these pose problems in this system. Individual male T. castaneum can mate with multiple females throughout their long life, successfully inseminating an average of 50 females within 7 days if given the opportunity (Lumley et al., 2015), so individual ejaculate sperm number/volume/concentration will not reveal the capacity for overall sperm number production. (If the reviewer refers to sperm cell volume for consideration, T. castaneum sperm cells are also small and extremely thin, and may vary in diameter along their long axis, so volume will also be a challenge to measure accurately.) Finally, and as in most insects, motility is also a problem to measure meaningfully in T. castaneum because sperm evidently function through the female tract by moving within narrow tubules or within dense sperm masses, so the motility of individual isolated sperm (even if the tract biochemistry could be replicated) on a microscope slide will likely not reflect function in the natural fertilisation environment.

We therefore now suggest the following additional interpretations for why the 38^o^C thermal selection lines have evolved slightly longer sperm, whereas short-term temperature increases through pupal and adult development cause a reduction in sperm length:

**“**Although we found evidence that laboratory selection across 54 generations at 38°C caused the overall evolution of ~4% longer sperm compared with the thermal line selection at 30°C, our within-generation experiments revealed consistent ~7% reductions in sperm length for all populations when produced through the warmer 38°C regime compared with 30°C (Figure 2). […] Whatever the cause behind long-term experimental evolution of sperm length within our thermal lines, it is clear that short-term within-generation impacts of temperature have strong and direct effects on sperm development, with an experimental switch to warmer regimes for either the thermal selection lines or the ancestral stock population resulting in consistently clear reductions in sperm length (Figure 2). Temperature variation during gametogenesis is known to affect sperm size in ectotherms…”

The authors need to justify the experimental temperatures the beetles were exposed to. In special, 38 C is not a yearly average temperature that exists anywhere in the world. Even the lower temperature (30 C) is not a common average temperature anywhere in the world. And very unlikely it will exist, even under global warming scenarios. So, the paper can be better framed in the context of seasonal temperature variability (growing seasons, summer, etc.) and not necessarily in a global warming context. Comparing only two temperatures 30 vs. 38 C is not ideal, besides being temperatures that are very far apart from each other, it is impossible to draw a general trend when you only have one comparison. It would have been great had the manuscript had an intermediate temperature treatment, at 34 C (or so).

Indeed, we agree that our experimental thermal exposures are not relevant to annual average temperature effects, but seasonal variation, which is becoming more extreme as climate change drives up atmospheric volatility. We now frame the work more in this context, and make it clearer throughout the manuscript that our study is relevant to thermal variability, and not annual averages (including three new references to this). Although we cite climate change as being relevant to our study, we believe that our findings have primary impact in revealing a novel example of basic biological function for dealing with inherent thermal variation in the natural environment. Temperatures exceeding 38°C have been recorded in 150 countries (Mherrera, 2019), so are very frequently encountered in tropical latitudes. It is also relevant to note that the laboratory ‘optimum’ for population productivity in our T. castaneum model is a warm 35°C (stated in the manuscript Materials and methods), so 38°C is not too far apart from that.

We also agree that additional thermal regimes in the study would have been great, but the existing workload to complete this work through a number of years at control and extreme (but relevant) experimental temperatures was itself more than a stretch.

Changes to framing the work:

“One of the most important abiotic environmental variables is temperature (Cossins and Bowler 1987;Angilletta Jr., 2009), especially in the context of climate change when thermal environments are expected to both warm and become much variable and extreme (Perkins et al., 2012, Raftery et al., 2017).”

**“**After exposing adult males and females and their gametes to different temperatures, we compare sperm and egg development and reproductive function within thermal regimes that mimic the increasingly variable conditions faced by warmer tropical regions (Perkins et al., 2012).”

“…an obvious mechanism to enable adaptive thermal plasticity for populations facing the challenge of reproducing under climate change where increases in both thermal averages, maxima and variation are expected (Perkins et al., 2012, Raftery et al., 2017).”

**“**30°C is our standard rearing temperature, and the optimum laboratory temperature for population productivity in *T. castaneum* is 35°C (Sokoloff, 1974, Sales, 2019). […] Our experimental exposure to this thermal regime will have relevance to more extreme thermal conditions and heatwaves, predicted to increase in frequency, severity and duration under climate change (e.g. Perkins et al., 2012), and which we know specifically constrain reproduction and sperm function in this system (Sales et al., 2018).”